# Copper-catalyzed enantioselective Sonogashira-type oxidative cross-coupling of unactivated C($sp^3$)−H bonds with alkynes

Zhen-Hua Zhang[1,2,4], Xiao-Yang Dong[2,4], Xuan-Yi Du[2,4], Qiang-Shuai Gu [3], Zhong-Liang Li[3] & Xin-Yuan Liu [2]*

Transition metal-catalyzed enantioselective Sonogashira-type oxidative C($sp^3$)—C($sp$) coupling of unactivated C($sp^3$)—H bonds with terminal alkynes has remained a prominent challenge. The difficulties mainly stem from the regiocontrol in unactivated C($sp^3$)—H bond functionalization and the inhibition of readily occurring Glaser homocoupling of terminal alkynes. Here, we report a copper/chiral cinchona alkaloid-based *N,N,P*-ligand catalyst for asymmetric oxidative cross-coupling of unactivated C($sp^3$)—H bonds with terminal alkynes in a highly regio-, chemo-, and enantioselective manner. The use of *N*-fluoroamide as a mild oxidant is essential to site-selectively generate alkyl radical species while efficiently avoiding Glaser homocoupling. This reaction accommodates a range of (hetero)aryl and alkyl alkynes; (hetero)benzylic and propargylic C($sp^3$)−H bonds are all applicable. This process allows expedient access to chiral alkynyl amides/aldehydes. More importantly, it also provides a versatile tool for the construction of chiral C($sp^3$)—C($sp$), C($sp^3$)—C($sp^2$), and C($sp^3$)—C($sp^3$) bonds when allied with follow-up transformations.

[1] Shandong Provincial Key Laboratory of Detection Technology for Tumor Markers, School of Chemistry and Chemical Engineering, Linyi University, Linyi 276005, China. [2] Shenzhen Grubbs Institute and Department of Chemistry, Southern University of Science and Technology, Shenzhen 518055, China. [3] Academy for Advanced Interdisciplinary Studies and Department of Chemistry, Southern University of Science and Technology, Shenzhen 518055, China. [4] These authors contributed equally: Zhen-Hua Zhang, Xiao-Yang Dong, Xuan-Yi Du. *email: liuxy3@sustech.edu.cn

As one of the most fundamental motifs in organic chemistry, chiral alkynes play an essential role in biology, medicinal chemistry, and material science[1,2]. They also serve as vital synthetic precursors for many functionalities such as alkanes, alkenes, aldehydes, carboxylic acids, and heterocycles on both laboratory and industrial scales[1,2]. Accordingly, a variety of catalytic methods have been developed to enantioselectively deliver chiral C($sp^3$)—C($sp$) bonds[3–13]. From the viewpoint of step- and atom-economy, a direct enantioselective alkynylation of a C($sp^3$)—H bond would be highly appealing[14–16]. Thus, directed palladium-catalyzed C—H activation has been shown to be an effective strategy for C($sp^3$)—H alkynylation[17–19]. However, only a few enantioselective examples have been disclosed with the expensive alkynyl bromide/iodine as the alkynylation reagents[16,20–22]. In comparison, an enantioselective Sonogashira-type cross-dehydrogenative coupling (CDC) of C($sp^3$)—H bonds with low-cost terminal alkynes would be more ideal due to the ready availability of both coupling partners. In this respect, Li and others have pioneered in establishing enantioselective alkynylation of C($sp^3$)—H bonds adjacent to nitrogen with terminal alkynes[23–32]. The stereochemical control was elegantly implemented via chiral Lewis acid-catalyzed nucleophilic addition to the in situ generated iminium ions (Fig. 1a)[23–32]. However, the enantioselective oxidative cross-coupling of common unactivated C($sp^3$)—H bonds poses a significant challenge to this strategy. Two major difficulties have to be overcome: (1) The site-selective generation of carbocation species from such unactivated C($sp^3$)—H bonds is relatively difficult compared with that adjacent to nitrogen. (2) The chemo- and enantiocontrol of the non-heteroatom-stabilized carbocations are also more challenging due to their inherently high reactivity. Clearly, a conceptually different approach is highly desirable to achieve the enantioselective oxidative cross-coupling of unactivated C($sp^3$)—H bonds and terminal alkynes.

Enantioselective C($sp^3$)—H bond functionalization by merging a site-selective hydrogen atom abstraction (HAA) process with asymmetric copper-catalyzed cross-coupling has received much attention over the past several years[33–42]. At the same time, we have recently developed a copper/cinchona alkaloid-based N,N-ligand catalyst, which could intimately associate with alkyl radical species for realization of asymmetric alkene difunctionalization[43–46]. Inspired by these works, we wondered if this catalyst system would be also suitable for tandem site-selective HAA on unactivated C($sp^3$)—H bonds and enantioselective coupling with terminal alkynes (Fig. 1b). However, common oxidants for HAA readily lead to copper-catalyzed Glaser homocoupling of terminal alkynes[47,48]. To this end, we have been encouraged by the mild oxidation power of N-halogenated amides employed in recent remote C($sp^3$)—H

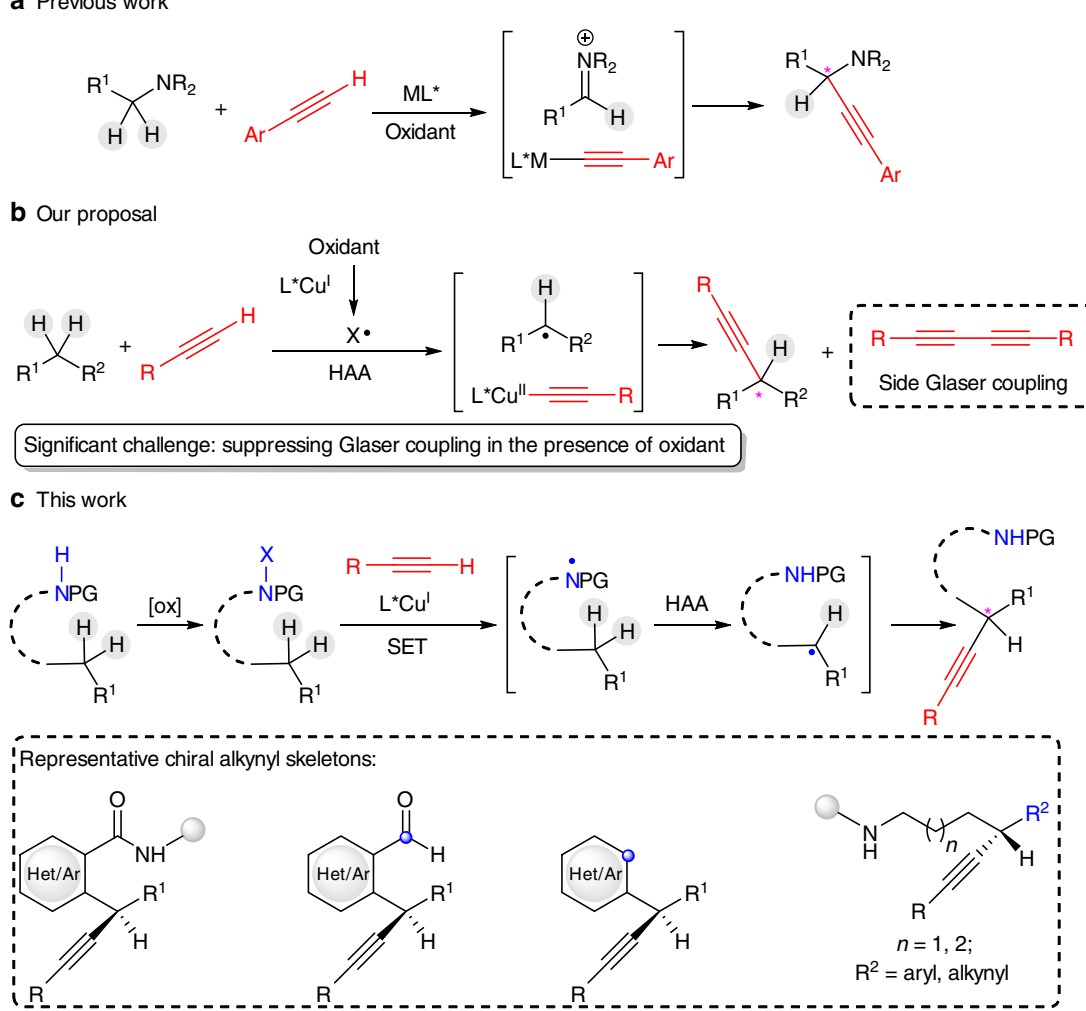

**Fig. 1 Sonogashira-type enantioselective oxidative cross-coupling of C($sp^3$)—H bonds with terminal alkynes. a** Previous ionic-type dehydrogenative coupling of C($sp^3$)—H bonds adjacent to nitrogen. **b** Our proposal: tandem HAA and copper-catalyzed Sonogashira-type coupling. **c** Amide-directed enantioselective coupling of C($sp^3$)—H bonds with terminal alkynes. HAA hydrogen atom abstraction, [ox] oxidant, SET single-electron transfer.

**Table 1 Screening of reaction conditions[a].**

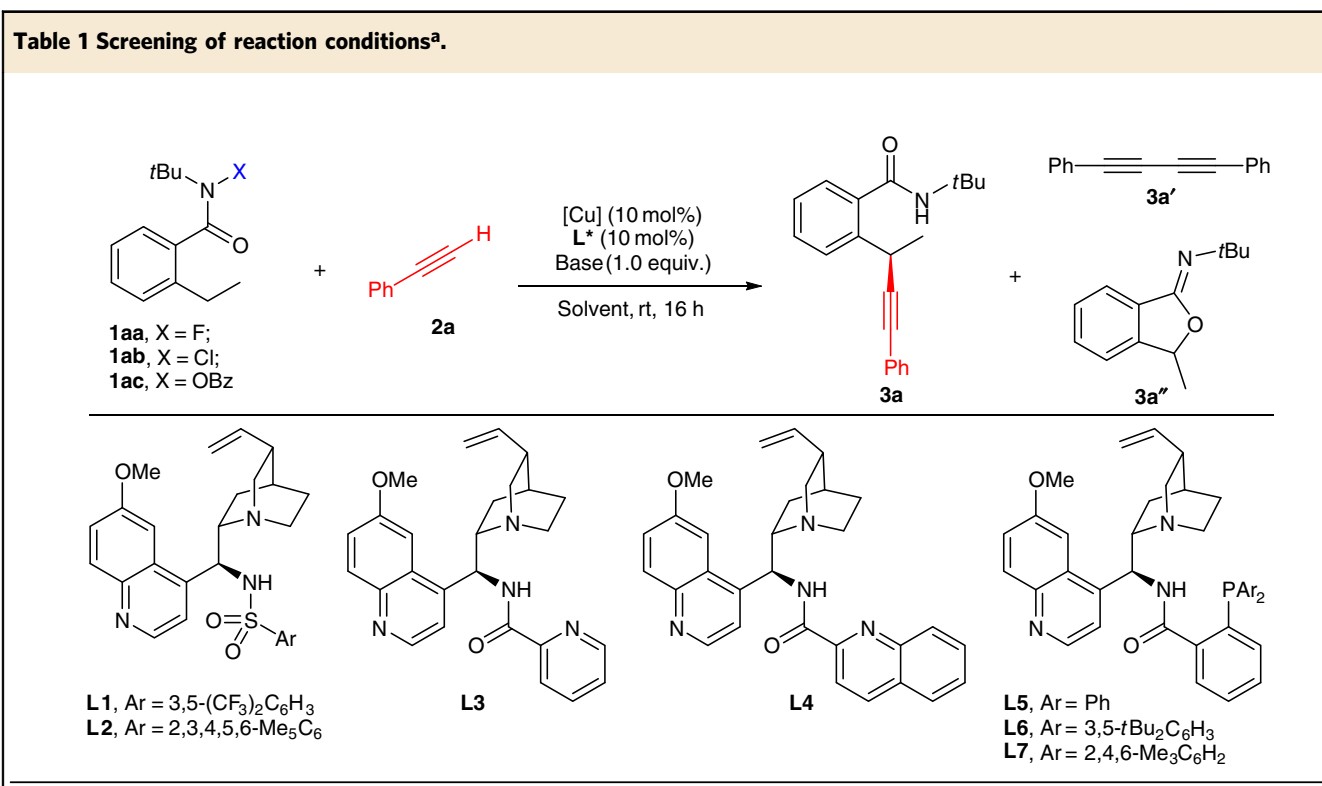

| Entry | 1 | [Cu] | Base | L* | Solvent | Yield (%, 3a, 3a′, 3a″)[b] | Ee (%)[c] |
|---|---|---|---|---|---|---|---|
| 1 | **1aa** | CuI | Cs$_2$CO$_3$ | **L1** | Dichloromethane (DCM) | –[d], 0, 0 | – |
| 2 | **1aa** | CuI | Cs$_2$CO$_3$ | **L2** | DCM | –[d], 0, 0 | – |
| 3 | **1aa** | CuI | Cs$_2$CO$_3$ | **L3** | DCM | <5, 0, 0 | 6 |
| 4 | **1aa** | CuI | Cs$_2$CO$_3$ | **L4** | DCM | 8, 0, 0 | 74 |
| 5 | **1aa** | CuI | Cs$_2$CO$_3$ | **L5** | DCM | 53, 0, –[d] | 68 |
| 6 | **1aa** | CuI | Cs$_2$CO$_3$ | **L6** | DCM | 15, 0, 0 | 51 |
| 7 | **1ab** | CuI | Cs$_2$CO$_3$ | **L5** | DCM | –[d], 48, 0 | – |
| 8 | **1ac** | CuI | Cs$_2$CO$_3$ | **L5** | DCM | 0, 0, –[d] | – |
| 9 | **1aa** | CuI | Cs$_2$CO$_3$ | **L5** | 1,2-Dichloroethane | 50, 0, –[d] | 76 |
| 10 | **1aa** | CuI | Cs$_2$CO$_3$ | **L5** | Benzene | 77, 0, 5 | 64 |
| 11 | **1aa** | CuI | Cs$_2$CO$_3$ | **L5** | EtOAc | 92, 0, 0 | 91 |
| 12 | **1aa** | CuI | Cs$_2$CO$_3$ | **L5** | THF | 88, 0, 0 | 94 |
| 13 | **1aa** | CuBr | Cs$_2$CO$_3$ | **L5** | THF | 64, 0, 0 | 93 |
| 14 | **1aa** | CuTc | Cs$_2$CO$_3$ | **L5** | THF | 91, 0, 0 | 92 |
| 15 | **1aa** | CuOAc | Cs$_2$CO$_3$ | **L5** | THF | 80, 0, 0, | 94 |
| 16 | **1aa** | CuI | Na$_2$CO$_3$ | **L5** | THF | <5, 0, 81 | 85 |
| 17 | **1aa** | CuI | K$_2$CO$_3$ | **L5** | THF | 27, 0, 62 | 93 |
| 18 | **1aa** | CuI | KO$t$Bu | **L5** | THF | 71, 0, 26 | 93 |
| 19[e] | **1aa** | CuI | Cs$_2$CO$_3$ | **L5** | THF | 66, 0, 0 | 94 |
| 20[f] | **1aa** | CuI | Cs$_2$CO$_3$ | **L5** | THF | 31, 0, 0 | 92 |
| 21 | **1aa** | CuI | Cs$_2$CO$_3$ | **-** | THF | 0, 0, 0 | – |
| 22[g] | **1aa** | CuI | Cs$_2$CO$_3$ | **-** | THF | 0, 0, 0 | – |
| 23[g] | **1aa** | CuI | Cs$_2$CO$_3$ | **L5** | THF | 0, 0, 0 | – |

[a]Reaction conditions: **1a** (0.1 mmol), **2a** (0.2 mmol), [Cu] (10 mol%), **L*** (10 mol%), and base (0.1 mmol) in dry solvent (1.2 mL) at room temperature (rt) for 16 h
[b]Yield based on $^1$H NMR analysis of the crude product using CH$_2$Br$_2$ as an internal standard
[c]Ee values based on HPLC analysis
[d]A trace amount of product
[e]CuI (5 mol%), **L*** (5 mol%) for 24 h
[f]CuI (2 mol%), **L*** (2 mol%) for 36 h
[g]Without **2a**

functionalization transformations[49–59] based on the classic Hofmann–Löffler–Freytag[60,61] and Barton[62] reactions. More importantly, the corresponding amidyl radicals[49–51] commonly exhibit robust HAA reactivity and site-selectivity. Thus, we questioned if a removable directing amide group would be viable for sequential *N*-oxidation and asymmetric copper-catalyzed Sonogashira-type coupling of unactivated C−H bonds with terminal alkynes.

Herein, we describe our efforts toward the development of radical asymmetric oxidative cross-coupling of unactivated C($sp^3$) −H bonds with terminal alkynes enabled by copper(I)/cinchona alkaloid-based *N,N,P*-ligand catalysis. Notably, this protocol not only provides a range of chiral alkynyl amides and alkynyl aldehydes (Fig. 1c) but also, together with further transformations, offers a general way for chiral C($sp^3$)–C($sp$), C($sp^3$)–C($sp^2$), and C($sp^3$)–C($sp^3$) bond construction.

## Results

**Reaction optimization.** To verify our hypothesis, *N*-halo-amides **1aa** and **1ab** as well as *O*-acylhydroxylamide **1ac** were prepared, which all can generate the amidyl radical via single-electron reduction by a Cu(I) catalyst[54,56]. Initial treatment of *N*-fluoroamide **1aa** with phenylacetylene **2a** and Cs$_2$CO$_3$ in the presence of CuI and our designed bidentate quinine-derived sulfonamide ligands[43] **L1** and **L2** did not provide any products (entries 1 and 2, Table 1). We tentatively ascribed the results to the insufficient reducing capability of Cu(I) catalysts. Thus, we speculated that an additional coordinative amine or phosphine moieties on the ligands might help increase the electron-density on copper and enhance its reducing power. As such, cinchona alkaloid-derived Nakamura's tridentate *N,N,N*-ligand[63] **L3** and Dixon's *N,N,P*-ligand[64] **L5** as well as their analogs **L4** and **L6**, respectively, were examined. And most of them led to the desired product **3a** with promising enantioselectivity (entries 3–6, Table 1). In contrast, *N*-chloroamide **1ab** and *O*-acylhydroxylamide **1ac** only afforded Glaser homocoupling product **3a′** (entries 7 and 8, Table 1) under such reaction conditions, indicating the importance of the chosen amidyl radical precursor. Further screening of Cu(I) catalysts and solvents (entries 9–15, Table 1) proved that CuI in Tetrahydrofuran (THF) was the best (88% yield and 94% ee; entry 12, Table 1). An evaluation of base indicated significant impacts on the efficiency and selectivity (entries 16–18, Table 1) and Cs$_2$CO$_3$ was found to be particularly effective to inhibit the formation of side product **3a″**. Lowering the catalyst loading affected the yield but not the enantioselectivity (entries 19 and 20, Table 1). Control experiments showed that both the *N,N,P*-ligand and the alkyne were necessary for efficient amide consumption (entries 21–23, Table 1). Thus, both of them are required for the copper center to efficiently reduce the mild N–F oxidant.

**Substrate scope.** With the optimized conditions in hand, the scope of alkynes was next explored (Fig. 2). A series of aryl

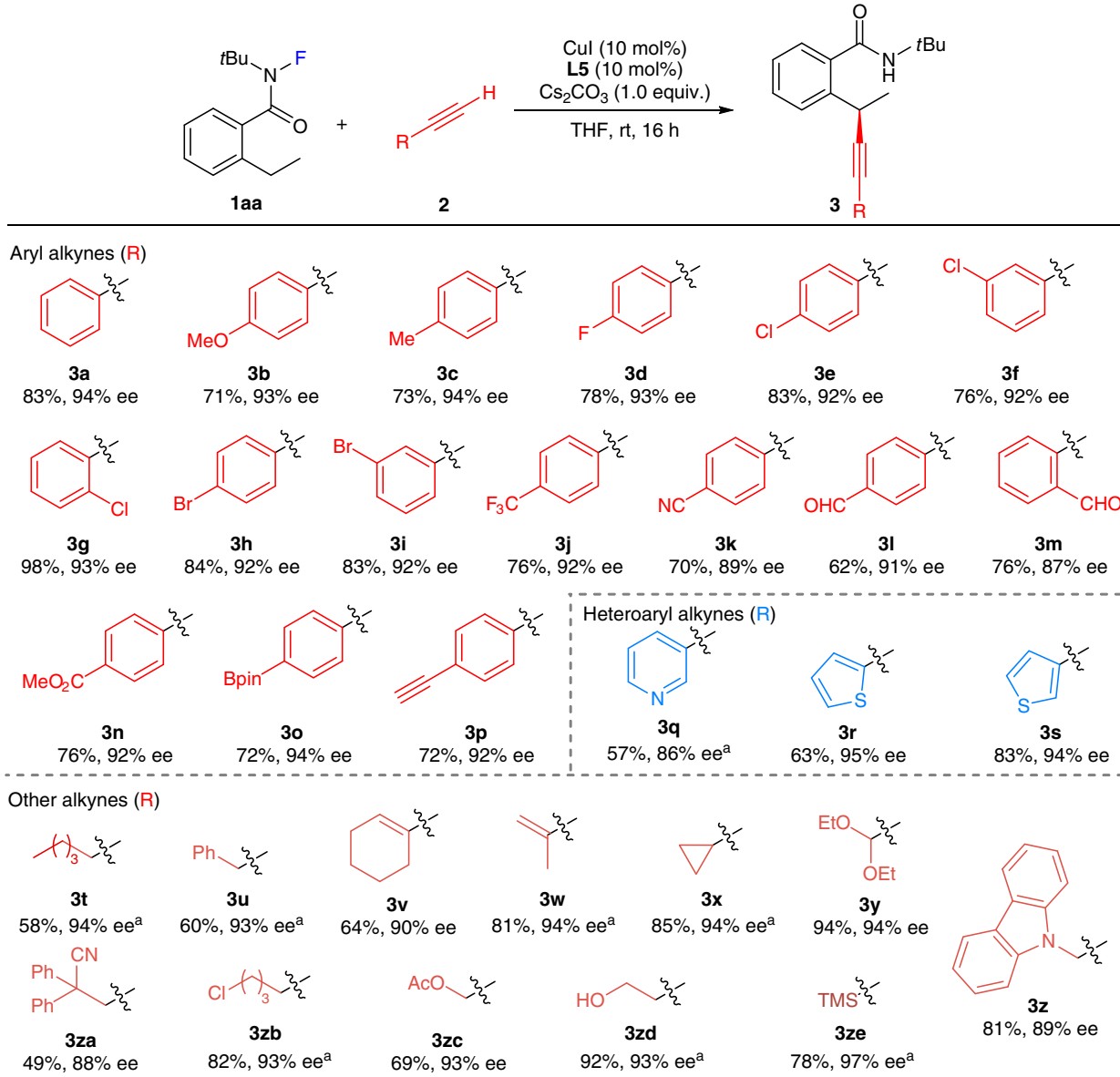

**Fig. 2 Substrate scope of alkynes.** Standard conditions: **1aa** (0.2 mmol), alkyne (0.4 mmol), CuI (10 mol%), **L5** (10 mol%), and Cs$_2$CO$_3$ (1.0 equiv.) in THF (2.4 mL) at rt for 16 h. Isolated yield based on **1aa** is given. Ee values are based on HPLC analysis. [a]CuTc (15 mol%), **L5** (10 mol%), and Cs$_2$CO$_3$ (2.0 equiv.) in THF at rt for 24 h. Bpin pinacolborato, TMS trimethylsilyl.

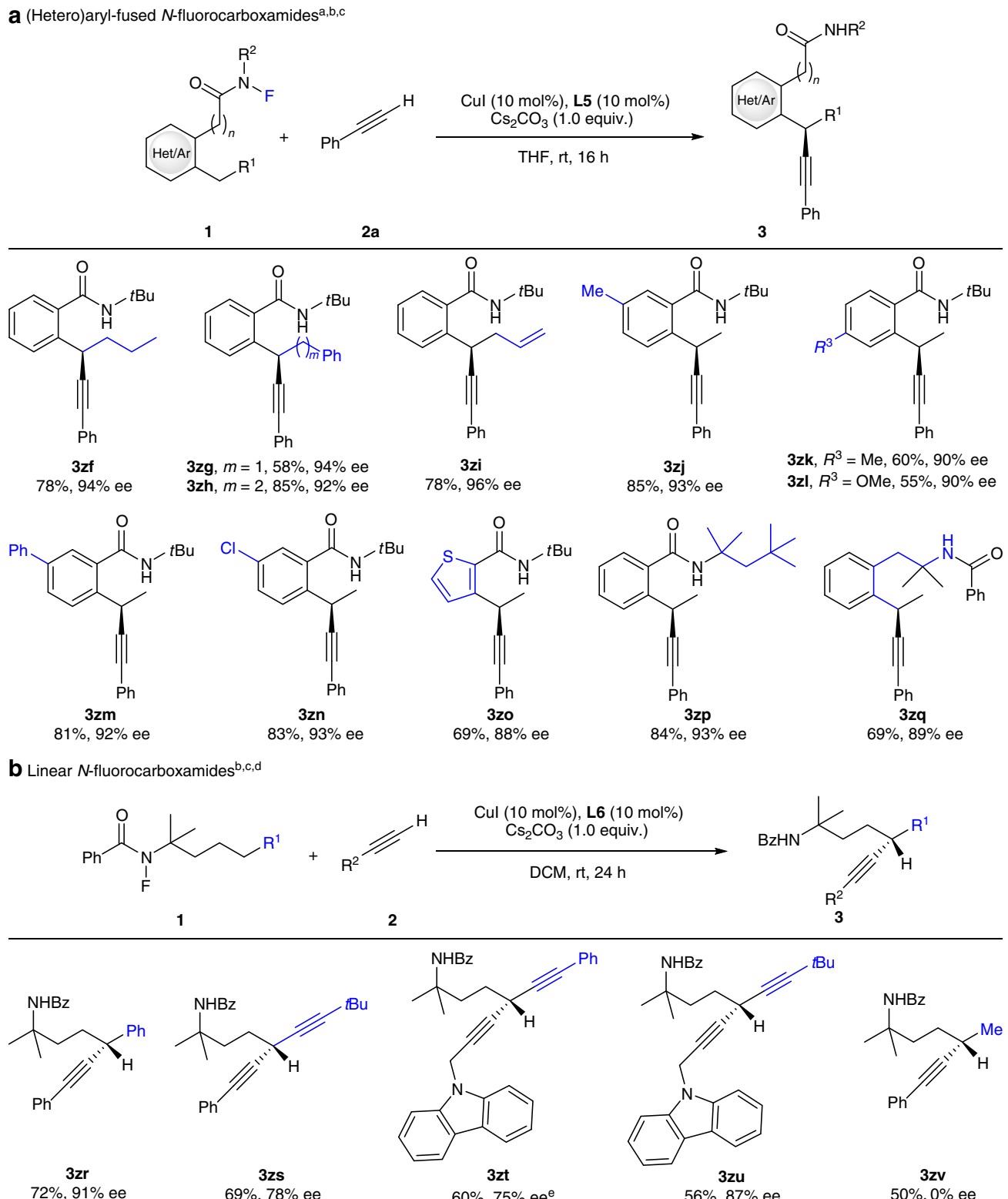

**Fig. 3 Substrate scope of *N*-fluorocarboxamides. a** The reaction is compatible with a variety of (hetero)aryl-fused *N*-fluorocarboxamide substrates. **b** Linear *N*-fluorocarboxamides are also applicable under slightly modified conditions. [a]Standard conditions: **1** (0.2 mmol), alkyne (0.4 mmol), CuI (10 mol%), **L5** (10 mol%), and Cs$_2$CO$_3$ (1.0 equiv.) in THF (2.4 mL) at rt for 16 h. [b]Isolated yield based on **1**. [c]Ee values based on HPLC analysis. [d]**L6** (10 mol%) was used in DCM at rt for 24 h. [e]**L7** (Table 1, 10 mol%) was used in CHCl$_3$ at rt for 24 h.

alkynes, including those having electron-donating or -withdrawing groups at different positions (*ortho*, *meta*, or *para*) of phenyl rings, reacted smoothly to afford **3a–3p** in 62–98% yield with 87–94% ee. Many functional groups, such as methoxyl (**3b**),

halo (**3d–3i**), trifluoromethyl (**3j**), cyano (**3k**), formyl (**3l** and **3m**), methoxylcarbonyl (**3n**), pinacolborato (**3o**), and terminal alkynyl (**3p**), were all compatible with the reaction conditions. Furthermore, a range of heteroaryl alkynes, such as 3-pyridinyl,

2-thiophenyl, and 3-thiophenyl, all worked well to deliver **3q**–**3s** in good yields with excellent enantioselectivity. We were especially pleased to find that alkyl alkynes were also competent coupling partners. For example, the aliphatic alkynes underwent the reaction to give **3t** and **3u** with good results in the presence of 2 equivalents of $Cs_2CO_3$. A wide range of functional groups, such as conjugating alkenyl (**3v** and **3w**), cyclopropanyl (**3x**), acetal (**3y**), carbazole (**3z**), cyano (**3za**), primary chloride (**3zb**), ester (**3zc**), and even hydroxy (**3zd**) groups, at different distances away from the reacting alkynes were well tolerated, therefore giving products in excellent enantioselectivity. In addition, a silyl alkyne was also applicable to provide **3ze** in 78% yield with 97% ee.

The scope of N-fluorocarboxamides bearing secondary benzylic $C(sp^3)$−H bonds was next evaluated (Fig. 3a). Simple alkyl-substituted substrates worked well to give alkynylation products **3zf**–**3zh** with excellent regio- and enantioselectivity. Noteworthy is that a terminal alkenyl group was tolerated under the reaction conditions to provide **3zi** in 78% yield with 96% ee. A range of electron-donating or -withdrawing substituents at different positions of the backbone phenyl rings were well accommodated to deliver **3zj**–**3zn** in 55–85% yield with 90–93% ee. In addition, direct alkynylation of the α-C−H bond of a thiophene heterocycle also proceeded smoothly to give **3zo** in 69% yield with 88% ee. The N-tBu group could be replaced with another tertiary alkyl group without any interferences on either yield or

enantioselectivity (**3zp**). The above method could also be extended to more remote benzylic C−H bonds via a 1,6-HAA[56,65–67] process to give alkynylated amide **3zq** in 69% yield with 89% ee.

To further test the compatibility of the reaction in more structurally diverse contexts, an N-benzoylated amine with a tethered phenyl ring was evaluated and the desired product **3zr** was obtained in 72% yield and 91% ee (Fig. 3b). Noteworthy is that the reaction is not restricted to benzylic or heterobenzylic $C(sp^3)$−H bond functionalization. For example, various propargylic $C(sp^3)$−H bonds were amenable to this transformation, therefore providing **3zs**−**3zu** in moderate to high enantioselectivity. Interestingly, a substrate containing only simple alkyl δ-C−H bonds also underwent the reaction to deliver **3zv** in excellent regioselectivity, albeit with no enantioselectivity. The reaction is currently under further optimization for potential enantiocontrol in our laboratory.

**Straightforward transformation.** One feature of this strategy is the requirement for an amide directing group. In fact, the amide group in product **3a** was readily removed by sequential treatment with the Schwartz's reagent and the Wilkinson's catalyst in two steps (Fig. 4a). In this sense, this protocol provides an indirect approach for enantioselective Sonogashira-type oxidative coupling. The absolute configuration of **5** was determined to be R by

**a** Removal of the amide group

**3a**, 92% ee **4**, 76%, 89% ee **5**, 70%, 89% ee

**b** Versatile follow-up transformations for the construction of chiral $C(sp^3)$–$C(sp^2)$ and $C(sp^3)$–$C(sp^3)$ bonds

**6** 90%, 89% ee **3a**, 92% ee **7** 96%, 88% ee

**c** Synthesis of chiral terminal alkyne building blocks

**3ze**, 97% ee **8**, 86%, 97% ee

**Fig. 4 Straightforward transformation. a** The directing amide group was readily removed by sequential amide reduction to aldehyde and decarbonylation. **b** The essential alkyne moiety in the product was straightforwardly transformed into Z-alkene and alkane featuring chiral $C(sp^3)$–$C(sp^2)$ and $C(sp^3)$–$C(sp^3)$ bonds. **c** Silyl alkyne was easily converted to terminal alkyne, thus providing valuable chiral building blocks.

comparing its HPLC spectrum and optical rotation with those reported in literature[68,69]. Another feature of this protocol is the reactive alkyne moieties in products, which could be readily converted to a Z-alkene group in **6** and a saturated alkyl group in **7** upon hydrogenation to different extents (Fig. 4b), respectively. Therefore, it provides a versatile and complementary tool to other direct methods for the construction of chiral C($sp^3$)–C($sp^2$) and C($sp^3$)–C($sp^3$) bonds. In addition, the TMS group in **3ze** could be straightforwardly unmasked to provide the chiral terminal alkyne **8** in 86% yield without any loss of enantioselectivity (Fig. 4c). Such enantioenriched terminal alkynes are valuable chiral building blocks for a range of transformations.

**Mechanism investigation**. To gain some insight into the reaction mechanism, control experiments with radical inhibitors 2,2,6,6-tetramethyl-1-piperidinyloxy (TEMPO) and butylated

hydroxytoluene (BHT) were conducted, respectively, indicating reaction inhibition (Supplementary Fig. 1a). Moreover, a radical clock experiment showed that substrate **9** underwent a tandem cyclopropane ring-opening/alkyne trapping process to provide **10** in 72% yield with 74% ee (Fig. 5a). These observations support the possible formation of alkyl radical species from in situ-generated amidyl radical species via 1,5(6)-HAA processes. Deuterium-labeling experiments indicated an intramolecular kinetic isotope effect (KIE) value of 1.94 and an intermolecular KIE value of 1.16. Thus, the 1,5-HAA process might be not involved in the rate-determining step(s) (Supplementary Fig. 1b and 1c)[70]. In addition, only in the presence of **L5** could copper phenylacetylide initiate the reaction (Fig. 5b). This result, together with the aforementioned effects of ligand and phenylacetylene for efficient amide consumption (entries 21–23, Table 1), demonstrates that the complex of copper phenylacetylide with the chiral ligand might be the initial complex to start this reaction.

**Fig. 5 Mechanistic investigations. a** The radical-clock substrate (±)-**9** underwent ring opening before the C–C bond formation, thus indicating the initial generation of a benzylic radical. **b** The reaction of **1aa** with copper acetylide did not occur in the absence of **L5**, indicating that both chiral ligand and terminal alkyne are indispensable for reaction initiation. **c** The reaction was proposed to proceed through sequential single-electron reduction of substrate **1**, 1,5(6)-HAA, and copper-catalyzed C($sp^3$)–C($sp$) coupling.

On the basis of abovementioned observations and previous reports[38–46,52–59], a plausible mechanism was tentatively proposed, as shown in Fig. 5c. Initially, $Cu^IX$ reacts with chiral ligand and terminal alkyne in the presence of base, giving chiral Cu(I) acetylide complex **B**. Subsequent single-electron transfer (SET) of **B** with N-fluorocarboxamide **1** results in the formation of Cu(II) acetylide complex **C** and the amidyl radical **D**. The amidyl radical then undergoes intramolecular 1,5(6)-HAA to generate alkyl radical species **E**. Next, $C(sp^3)$–$C(sp)$ coupling via reductive elimination of a Cu(III) intermediate **F**[33–42] gives rise to enantioenriched product **3** and chiral Cu(I) complex **A**.

## Discussion

We have discovered copper/cinchona alkaloid-based N,N,P-ligand catalysts to accomplish a radical asymmetric oxidative $C(sp^3)$–$C(sp)$ cross-coupling of unactivated $C(sp^3)$−H bonds and terminal alkynes. The utilization of a removable amide group to direct the site-selective formation of alkyl radical species via a HAA process is crucial to the success of the transformation. In addition, the use of N-fluorocarboxamides as mild amidyl radical precursors is critical for inhibiting the Glaser homocoupling. Further, the strategic utilization of chiral cinchona alkaloid-derived N,N,P-ligands proved to be essential for eliciting the initial SET between copper and the mild N-fluorocarboxamides oxidant while imparting excellent enantiodiscrimination in the final C–C coupling step. This strategy allows for the facile assembly of chiral alkynyl amides/aldehydes and also provides a generally robust tool for the construction of chiral $C(sp^3)$–$C(sp)$, $C(sp^3)$–$C(sp^2)$, and $C(sp^3)$–$C(sp^3)$ bonds. Further studies toward the development of direct enantioselective oxidative Sonogashira-type coupling of unactivated $C(sp^3)$−H bonds with terminal alkynes are ongoing in our laboratory.

## Methods

**General procedure A**. This procedure applies to compounds **3a–p, 3r, 3s, 3v, 3y–za, 3zc, 3zf–zq**. Under argon atmosphere, an oven-dried resealable Schlenk tube equipped with a magnetic stir bar was charged with CuI (3.8 mg, 0.020 mmol, 10 mol%), **L5** (12.2 mg, 0.020 mmol, 10 mol%), $Cs_2CO_3$ (65.2 mg, 0.20 mmol, 1.0 equiv.), and anhydrous THF (2.4 mL). Then, N-fluorocarbox-amide (0.20 mmol, 1.0 equiv.) and alkyne (0.40 mmol, 2.0 equiv.) were sequentially added into the mixture and the reaction mixture was stirred at (rt) for 16 h. Upon completion (monitored by thin-layer chromatography (TLC)), the precipitate was filtered off and washed by DCM. The filtrate was evaporated and the residue was purified by column chromatography on silica gel to afford the desired product.

**General procedure B**. This procedure applies to compounds **3q, 3t, 3u, 3w, 3x, 3zb, 3zd, 3ze**. Under argon atmosphere, an oven-dried resealable Schlenk tube equipped with a magnetic stir bar was charged with CuTc (5.7 mg, 0.030 mmol, 15 mol%), **L5** (12.2 mg, 0.020 mmol, 10 mol%), $Cs_2CO_3$ (130.4 mg, 0.40 mmol, 2.0 equiv.), and anhydrous THF (2.4 mL). Then, N-fluorocarboxamide (0.20 mmol, 1.0 equiv.) and alkyne (0.40 mmol, 2.0 equiv.) were sequentially added into the mixture and the reaction mixture was stirred at rt for 24 h. Upon completion (monitored by TLC), the precipitate was filtered off and washed by DCM. The filtrate was evaporated and the residue was purified by column chromatography on silica gel to afford the desired product.

**General procedure C**. This procedure applies to compounds **3zr, 3zs, 3zu, 3v**. Under argon atmosphere, an oven-dried resealable Schlenk tube equipped with a magnetic stir bar was charged with CuI (3.8 mg, 0.020 mmol, 10 mol%), **L6** (16.7 mg, 0.020 mmol, 10 mol%), $Cs_2CO_3$ (65.2 mg, 0.20 mmol, 1.0 equiv.), and anhydrous DCM (2.4 mL). Then, N-fluorocarboxamide (0.20 mmol, 1.0 equiv.) and alkyne (0.40 mmol, 2.0 equiv.) were sequentially added into the mixture and the reaction mixture was stirred at rt for 24 h. Upon completion (monitored by TLC), the precipitate was filtered off and washed by DCM. The filtrate was evaporated and the residue was purified by column chromatography on silica gel to afford the desired product.

**General procedure D**. This procedure applies to compound **3zt**. Under argon atmosphere, an oven-dried resealable Schlenk tube equipped with a magnetic stir bar was charged with CuI (3.8 mg, 0.020 mmol, 10 mol%), **L7** (13.9 mg,

0.020 mmol, 10 mol%), $Cs_2CO_3$ (65.2 mg, 0.20 mmol, 1.0 equiv.), and anhydrous chloroform (2.4 mL). Then, N-fluorocarboxamide (0.20 mmol, 1.0 equiv.) and alkyne (0.40 mmol, 2.0 equiv.) were sequentially added into the mixture and the reaction mixture was stirred at rt for 24 h. Upon completion (monitored by TLC), the precipitate was filtered off and washed by DCM. The filtrate was evaporated and the residue was purified by column chromatography on silica gel to afford the desired product.

For nuclear magnetic resonance and high-performance liquid chromatography spectra, see Supplementary Figures.

## Data availability

Experimental procedure and characterization data of new compounds are available within Supplementary Information. Any further relevant data are available from the authors upon reasonable request.

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

## Acknowledgements

Financial support from the National Natural Science Foundation of China (Nos. 21722203, 21831002, 21801116, and 21804066), Shenzhen special funds for the development of biomedicine, internet, new energy, and new material industries (No. JCYJ 20180302174416591), Shenzhen Nobel Prize Scientists Laboratory Project (No. C17783101), and Shandong Provincial Natural Science Foundation (No. ZR2017BB065) is appreciated.

## Author contributions

Z.-H.Z., X.-Y.Do, and X.-Y.Du. performed the experiments. Q.-S.G. and Z.-L.L. helped with characterizing some new compounds. X.-Y.L. conceived and directed the project and wrote the paper. All authors discussed the results and commented on the manuscript.

## Competing interests

The authors declare no competing interests.
