## [Peer Review File · Nature Communications]

Reviewers' comments:

Reviewer #1 (Remarks to the Author):

In this manuscript, Liu and coworkers described a novel copper catalyst for the challenging asymmetric Sonogashira-type alkynylation of C(sp³)-H bonds via radical intermediates. The asymmetric C(sp³)-H functionalization via sequential hydrogen atom abstraction and copper-catalyzed coupling has recently emerged as a robust and efficient strategy. And known recent examples include asymmetric cyanation and arylation of C(sp³)-H bonds (Science 2016, 353, 1014-1018; Angew. Chem. Int. Ed. 2019, 58, 6425-6429). This work represents an intriguing example in this area since all those previous reactions have been based on the traditional chiral copper-bisoxazoline catalysts. On the other hand, asymmetric Sonogashira-type C(sp³)-C(sp) coupling from C(sp³)-H bonds provides an excellent approach for chiral C-C bond formation, which is one of the long-lasting main tasks in organic synthesis. In this sense, this work bears sufficient conceptual novelty and application significance for publication in Nature Communications. Nonetheless, some minor revisions are necessary, as listed below.

- 1) L1 in Table 1 is not corrected, should be Ar = 3,5-(CF₃)₂C₆H₃.
- 2) The "pinacolborato (3p), and terminal alkynyl (3p)," should be corrected.
- 3) What's the meaning of * in the "57%, 86% ee* of 3-pyridinyl 3q".
- 4) The 3ze in Table 2 is different from the 3ze in Fig. 3.
- 5) "(Hetero)aryl-tethered N-fluorocarboxamides" in Fig. 2 is not accurate.
- 6) Unify the format of C(spⁿ)-C(spⁿ).
- 7) The "89%ee" and "88%ee" in Fig. 3b should read as "89% ee" and "88% ee", respectively.
- 8) The tone concerning "therefore providing 4o-4q in high enantioselectivity" on Page 10 is not accurate since only ee values lower than 80% were obtained on products 4o and 4p.
- 9) The numbers for products should be adjusted so that structurally related products are uniformly numbered.
- 10) The "Methods" section should be revised so that different conditions for products 3a-3ze, 4b-4m, and 4n-4r are clearly described in detail, respectively.

Reviewer #2 (Remarks to the Author):

This paper describes an enantioselective cross-coupling of unactivated C-H bonds with alkynes. It builds on previous work involving alkynylation of C-H bonds adjacent to nitrogen. The paper is very clearly written, easy to follow and contains appropriate introduction, results and discussion sections.

This work is hugely impressive. The product yields and ee's (after optimization studies) are typically excellent, and the workers have taken the time to demonstrate the generality of the methodology (many examples are included). A sensible plausible mechanism is proposed and a comprehensive list of appropriate references is included. There are just a few very minor typographical errors (e.g. sp not always in italic; et. al for some references), but otherwise, I recommend publication.

This novel work will be of interest to a wide audience, and I enjoyed the opportunity to comment on such impressive work.

Andrew Parsons

Reviewer #3 (Remarks to the Author):

In this manuscript, Liu and coauthors realized an elegant example of asymmetric Sonogashira-type oxidative cross-coupling of unactivated C(sp³)-H bonds with terminal alkynes in the presence of copper/cinchona alkaloid-based N,N,P-ligand catalysts. The design of the reaction is very clever.

The utilization of a removable amide group to direct the site-selective formation of alkyl radical species via a HAA process was crucial to the success of this interesting transformation. The substrate scope is quite wide, including a series of alkynes and (hetero)aryl-tethered N-fluorocarboxamides. Besides 1,5 HAA, the method could also be extended to more remote benzylic C-H bonds via a 1,6-HAA process. This strategy provides a powerful tool for the construction of chiral C(sp³)-C(sp), C(sp³)-C(sp²), and C(sp³)-C(sp³) bonds. Thus, Publication in Nature Communications is recommended after minor revision of manuscript.

1. Why Cs₂CO₃ is particularly effective for inhibiting the formation of side product 3a"?
2. For dialkyne substrate 3p, is it possible to get double cross-coupling product by using 2 equivalents of 1aa?
3. In page 3, "N,N-ligand", the "N,N" should be italic. In page 4 and 5, "N,N,P-ligand", "N,N,P" should be italic. In page 10, C(sp³)-H, "sp³" should be italic.
4. An example of copper/guanidine-catalyzed asymmetric alkynylation of isatins was missing (Angew. Chem. Int. Ed. 2016, 55, 5286–5289.)
5. SI: High-resolution data (HRMS) of N-alkylbenzamides S-1m–S-1q should be added. The HRMS of compounds (3e, 3f, 3g, 3h, 3f, 3i, 3zb, 4j) contain Cl and Br should be given isotopic mass, and the character of catalytic products should be described in SI, such as 'colorless oil'.

Our Responses to the Comments of the Reviewers

Reviewer 1

Comment 1: *Liu and coworkers described a novel copper catalyst for the challenging asymmetric Sonogashira-type alkynylation of C(sp³)-H bonds via radical intermediates. The asymmetric C(sp³)-H functionalization via sequential hydrogen atom abstraction and copper-catalyzed coupling has recently emerged as a robust and efficient strategy. And known recent examples include asymmetric cyanation and arylation of C(sp³)-H bonds (Science 2016, 353, 1014-1018; Angew. Chem. Int. Ed. 2019, 58, 6425–6429). This work represents an intriguing example in this area since all those previous reactions have been based on the traditional chiral copper-bisoxazoline catalysts. On the other hand, asymmetric Sonogashira-type C(sp³)-C(sp) coupling from C(sp³)-H bonds provides an excellent approach for chiral C-C bond formation, which is one of the long-lasting main tasks in organic synthesis. In this sense, this work bears sufficient conceptual novelty and application significance for publication in Nature Communications.*

Our Response: We sincerely thank the reviewer for acknowledging the significance and novelty of this work and recommending its potential publication in *Nature Communications*.

Comment 2: *L1 in Table 1 is not corrected, should be Ar = 3,5-(CF₃)₂C₆H₃.*

Our Response: We are very grateful to the reviewer for pointing out the above error and feel very sorry for it. Accordingly, we have changed the word to “3,5-(CF₃)₂C₆H₃” in the revised manuscript, as suggested by this reviewer. We have also carefully revised the related text in the manuscript.

Comment 3: *The “pinacolborato (3p), and terminal alkynyl (3p),” should be corrected.*

Our Response: We are very grateful to the reviewer for pointing out the above error and feel very sorry for it. Accordingly, we have changed the text to “pinacolborato (**3o**), and terminal alkynyl (**3p**),” in the revised manuscript. We have also carefully revised the related text in the manuscript.

Comment 4: *What’s the meaning of * in the “57%, 86% ee* of 3-pyridinyl 3q”.*

Our Response: We sincerely thank this reviewer for bring this issue to our attention. We feel very sorry for it since it is a typo. Accordingly, we have changed the text to “57%, 86% ee^d” in the revised manuscript.

Comment 5: *The 3ze in Table 2 is different from the 3ze in Fig. 3.*

Our Response: We sincerely thank this reviewer for bring this issue to our attention and feel very sorry for this mistake. Accordingly, we have corrected the structure of **3ze** in Table 2 in the revised manuscript.

Comment 6: *“(Hetero)aryl-tethered N-fluorocarboxamides” in Fig. 2 is not accurate.*

Our Response: We sincerely thank this reviewer for bring this issue to our attention. Accordingly, we have changed “(Hetero)aryl-tethered N-fluorocarboxamides” to “**Linear** N-fluorocarboxamides” in Fig. 2 in the revised manuscript. We have also carefully revised the related text in the manuscript.

Comment 7: *Unify the format of C(spⁿ)-C(spⁿ).*

Our Response: We sincerely thank this reviewer for bring this issue to our attention. Accordingly, we have unified the format of **C(spⁿ)** throughout the revised manuscript and supplementary information.

Comment 8: *The “89%ee” and “88%ee” in Fig. 3b should read as “89% ee” and “88% ee”, respectively.*

Our Response: We are very grateful to the reviewer for pointing out the above errors and feel very sorry for them. Accordingly, we have changed these phrases to “**89% ee**” and “**88% ee**”, respectively, in the revised manuscript. We have also carefully revised the related text in the manuscript.

Comment 9: *The tone concerning “therefore providing 4o-4q in high enantioselectivity” on Page 10 is not accurate since only ee values lower than 80% were obtained on products 4o and 4p.*

Our Response: We sincerely appreciate the valuable suggestion by the reviewer. Accordingly, we have changed the text to “therefore providing **3zs–3zu** in **moderate to** high enantioselectivity” in the revised manuscript.

Comment 10: *The numbers for products should be adjusted so that structurally related products are uniformly numbered.*

Our Response: We sincerely thank this reviewer for bring this issue to our attention. Accordingly, we have adjusted the numbers for products to ensure the consistency of structurally related products in the revised manuscript and supplementary information. We have also carefully revised the related text in the manuscript and supplementary information.

Comment 11: *The “Methods” section should be revised so that different conditions for products 3a-3ze, 4b-4m, and 4n-4r are clearly described in detail, respectively.*

Our Response: We sincerely thank this reviewer for bring this issue to our attention. Accordingly, in order to clarify the reaction conditions, we have divided the reaction methods into four different categories in the revised manuscript.

Reviewer 2

Comment 1: *This paper describes an enantioselective cross-coupling of unactivated C–H bonds with alkynes. It builds on previous work involving alkynylation of C–H bonds adjacent to nitrogen. The paper is very clearly written, easy to follow and contains appropriate introduction, results and discussion sections. This work is hugely impressive. The product yields and ee's (after optimization studies) are typically excellent, and the workers have taken the time to demonstrate the generality of the methodology (many examples are included). A sensible plausible mechanism is proposed and a comprehensive list of appropriate references is included. There are just a few very minor typographical errors (e.g. *sp* not always in italic; *et. al* for some references), but otherwise, I recommend publication. This novel work will be of interest to a wide audience, and I enjoyed the opportunity to comment on such impressive work.*

Our Response: We sincerely thank the reviewer for the positive comments on our work and the recommendation for publication after appropriate revisions.

We also thank the reviewer for pointing out the typos to us and feel very for them. Accordingly, we have carefully rechecked the whole manuscript and corrected typos as many as possible in the revised manuscript. Specifically, we have formatted all “*sp*” in italic and used “*et al.*” all the time, as suggested by the reviewer.

Reviewer 3

Comment 1: *In this manuscript, Liu and coauthors realized an elegant example of asymmetric Sonogashira-type oxidative cross-coupling of unactivated C(sp³)-H bonds with terminal alkynes in the presence of copper/cinchona alkaloid-based N,N,P-ligand catalysts. The design of the reaction is very clever. The utilization of a removable amide group to direct the site-selective formation of alkyl radical species via a HAA process was crucial to the success of this interesting transformation. The substrate scope is quite wide, including a series of alkynes and (hetero)aryl-tethered N-fluorocarboxamides. Besides 1,5 HAA, the method could also be extended to more remote benzylic C-H bonds via a 1,6-HAA process. This strategy provides a powerful tool for the construction of chiral C(sp³)-C(sp), C(sp³)-C(sp²), and C(sp³)-C(sp³) bonds. Thus, Publication in Nature Communications is recommended after minor revision of manuscript.*

Our Response: We sincerely thank the reviewer for positively commenting on our work and recommending its publication in *Nature Communications*.

Comment 2: *Why Cs₂CO₃ is particularly effective for inhibiting the formation of side product 3a''?*

Our Response: We sincerely thank the reviewer for this valuable suggestion. The solubility and real base strength of Cs₂CO₃ are the highest among carbonates tested (Na₂CO₃, K₂CO₃, and Cs₂CO₃). Accordingly, the formation of copper acetylide and thus the Sonogashira coupling may be most facilitated. On the other hand, the obtained side product **3a''** is essentially racemic. Thus, the formation of **3a''** is likely via tandem oxidation of benzylic radical by Cu^{II} species to carbocation and electrophilic cyclization. The enhanced coordination of acetylide to copper may increase the electron density on the copper center, and thus, reduce its oxidation power. In this sense, the generation of benzylic carbocation and thus the formation of **3a''** may become disfavored. Nonetheless, further detailed experimental and theoretical studies are necessary to clearly delineate the mechanism, which are underway in our lab. The results will be disclosed in due course.

Comment 3: *For dialkyne substrate 3p, is it possible to get double cross-coupling product by using 2 equivalents of 1aa?*

Our Response: We sincerely thank the reviewer for this valuable suggestion. Accordingly, we have treated the dialkyne with 2.0 equivalents of *N*-fluoroamide **1aa** under our current optimal conditions and observed the double cross-coupling product in 44% yield with 87% ee and >20:1 dr. Therefore, the reaction can be readily switched for the single or double cross-coupling products, respectively, with high selectivity.

Scheme 1. Asymmetric double cross-coupling reaction of dialkyne with 1aa

2,2'-((2S,2'S)-1,4-phenylenebis(but-3-yn-4,2-diyl))bis(*N*-(*tert*-butyl)benzamide)

HPLC analysis: Chiralcel AD-H (*n*-hexane/*i*-PrOH = 80/20, flow rate 0.8 mL/min, $\lambda = 254$ nm), t_R (minor)

= 10.86 min, t_R (major) = 13.72 min.

$^1\text{H NMR}$ (400 MHz, CDCl_3): δ 7.73-7.66 (m, 2H), 7.42 (td, $J = 7.6, 1.5$ Hz, 2H), 7.32 (s, 6H), 7.28-7.23 (m, 2H), 5.74 (s, 2H), 4.49 (q, $J = 7.0$ Hz, 2H), 1.60 (d, $J = 7.0$ Hz, 6H), 1.46 (s, 18H); **$^{13}\text{C NMR}$** (100 MHz, CDCl_3): δ 169.2, 141.0, 136.5, 131.4, 130.1, 127.9, 126.8, 126.8, 123.0, 94.8, 81.7, 52.0, 29.0, 28.8, 24.0.

HRMS (ESI) m/z calcd. for $\text{C}_{36}\text{H}_{41}\text{N}_2\text{O}_2$ $[\text{M}+\text{H}]^+$ 533.3163, found 533.3162.

Signal 1: DAD1 A, Sig=254,4 Ref=360,100

Peak #	RetTime [min]	Type	Width [min]	Area [mAU*s]	Height [mAU]	Area %
1	10.779	BB	0.4485	2539.54004	89.11027	49.5689
2	13.687	BB	0.5347	2583.71338	74.85218	50.4311

Totals : 5123.25342 163.96245

Signal 1: DAD1 A, Sig=254,4 Ref=360,100

Peak #	RetTime [min]	Type	Width [min]	Area [mAU*s]	Height [mAU]	Area %
1	10.855	BB	0.4518	879.71936	30.56201	6.4634
2	13.721	MM R	0.5754	1.27311e4	368.75284	93.5366

Totals : 1.36108e4 399.31485

Comment 4: In page 3, “*N,N*-ligand”, the “*N,N*” should be italic. In page 4 and 5, “*N,N,P*-ligand”, “*N,N,P*” should be italic. In page 10, *C(sp³)-H*, “*sp³*” should be italic.

Our Response: We are very grateful to the reviewer for pointing out the above errors and feel very sorry for them. Accordingly, we have changed the phrases to “*N,N*-ligand”, “*N,N,P*-ligand”, and “*C(sp³)-H*” in the revised manuscript. We have also carefully revised the related text in the manuscript and supporting information.

Comment 5: An example of copper/guanidine-catalyzed asymmetric alkylation of isatins was missing (*Angew. Chem. Int. Ed.* 2016, 55, 5286–5289.)

Our Response: We greatly appreciate the reviewer’s reminding about this important reference and feel very sorry for overlooking it in the original manuscript. Accordingly, we have added this reference as ref. 3 in the revised manuscript.

Comment 5: *SI: High-resolution data (HRMS) of N-alkylbenzamides S-1m–S-1q should be added. The HRMS of compounds (3e, 3f, 3g, 3h, 3f, 3i, 3zb, 4j) contain Cl and Br should be given isotopic mass, and the character of catalytic products should be described in SI, such as ‘colorless oil’.*

Our Response: We sincerely thank this reviewer for these valuable suggestions. Accordingly, we have added HRMS data for all new compounds (**S-1b**, **S-1f–S-1q**, **[D₁]-S-1aa**, and **[D₂]-S-1aa**) in the revised supplementary information. We have also added the isotopic signals in the HRMS data of compounds containing Cl (**S-1j**, **1j**, **3e–3g**, **3zb**, and **3zn**) and Br (**3h** and **3i**), respectively. We have given descriptions of characters of all products in the revised supplementary information, too.

REVIEWERS' COMMENTS:

Reviewer #3 (Remarks to the Author):

The revised manuscript is suitable for publication in Nature Communications.

Point-by-Point Response to Reviewers

Reviewer 3#

Comment: *The revised manuscript is suitable for publication in Nature Communications.*

Our Response: We greatly appreciate the reviewer's recommendation of our work for publication in *Nature Communications*.